# Sharing or Hiding? Exploring the Influence of Social Cognition and Emotion on Employee Knowledge Behaviors within Enterprise Social Media

**DOI:** 10.3390/bs14080653

**Published:** 2024-07-28

**Authors:** Mingming He, Ziyi Yuan, Wenhao She

**Affiliations:** School of Economics & Management, Nanjing Tech University, Nanjing 211816, China; 202361113021@njtech.edu.cn (Z.Y.); 202361213228@njtech.edu.cn (W.S.)

**Keywords:** enterprise social media, social cognition, emotion, knowledge sharing, knowledge hiding

## Abstract

As emerging knowledge management platforms, enterprise social media (ESM) provide an important way for employees to engage in knowledge sharing and information communication within their organization. However, the question of how to encourage employees to engage in continuous knowledge sharing rather than knowledge hiding on ESM has not received sufficient attention from scholars. In contrast to previous studies that focused on a single theory perspective and a single knowledge behavior, in this study, we took a user cognition and emotion perspective and constructed a mechanism model for the impact of knowledge sharing and knowledge hiding among employees on ESM based on social cognition theory and emotion as social information theory. A total of 240 valid questionnaires were collected and used to empirically test the model. The results indicate that reciprocity and outcome expectancy have a significant positive effect on employees’ knowledge-sharing behavior, while reciprocity and trust have a significant negative effect on employees’ knowledge-hiding behavior. Positive emotions play a positive (enhancing) moderating role on the path between outcome expectancy and knowledge-sharing behavior, while negative emotions play a negative (weakening) moderating role on the path between reciprocity and knowledge-hiding behavior, as well as between trust and knowledge-hiding behavior. By incorporating employee emotions into the framework of social cognition’s impact on employee knowledge behavior, this study enriches theories related to enterprise social media, knowledge management, and user behaviors. Our research findings have practical implications for guiding employees to engage in positive knowledge sharing and reducing knowledge hiding on enterprise social media.

## 1. Introduction

In the digital era, an increasing number of businesses are adopting enterprise social media (ESM) to foster internal collaboration, communication, and knowledge exchange between employees [1,2,3,4]. Enterprise social media (ESM) represent an intuitive and cost-effective digital technology. They serve as internal social software within an enterprise, integrating various functions such as social networking, wikis, instant messaging, blogs, social bookmarking, and more [5,6]. They allow the employees of a company to communicate, collaborate, and interact with knowledge within that company by creating, editing, and commenting on content, with the potential to drive digital transformation strategies [1,2,7]. In particular, in the unique context of the global COVID-19 pandemic in 2020, ESM played a significant role in enabling businesses to combat the pandemic while maintaining normal production and operations successfully.

According to research conducted by McKinsey, the effective utilization of ESM can potentially increase employee productivity by 20–25%. Additionally, over 90% of Fortune 500 companies have implemented ESM technology into their daily business operations [8,9]. Relevant research reports indicate that the market for enterprise social applications, programs, and related services is projected to grow at a compound annual growth rate of 61% [10,11]. It is estimated that the investment in the social media market will reach USD 251.45 billion in 2024.

ESM offer technical capabilities such as visibility, editability, durability, and connectivity, which render them superior to traditional knowledge management systems and make them considered effective knowledge management tools [12,13,14,15]. However, in reality, the effectiveness of ESM in internal company usage is still not ideal. Research indicates that more than 80% of ESM fail to achieve the expected benefits and eventually become unsuccessful [16,17]. The author gained insights from a field survey conducted in several large IT companies, revealing that the efficiency of employees’ knowledge interactions on ESM remains relatively low. In fact, without a clear understanding of employees’ motivations and driving factors for knowledge interactions on ESM, it becomes challenging to understand their knowledge behaviors. This, in turn, impacts the efficiency of using ESM for internal knowledge exchange and collaboration among employees, hindering the achievement of the true value of ESM. Therefore, it is crucial to identify the antecedents of knowledge behavior on the ESM platforms from both theoretical and practical perspectives and develop appropriate incentive measures [18,19,20].

Scholars have extensively researched user knowledge behavior in online communities and public-orientated social media [21,22,23], examining various precedents, such as technology, organizational environment, user cognition, and motivation [18,20,24,25,26]. However, there is still a lack of research focusing on ESM specifically designed for internal use within organizations and for work-related purposes [4,27,28,29]. A few studies on enterprise social media have also focused on specific platforms, such as enterprise blogs, forums, and wikis [16,30]. They have examined the definition and characteristics of enterprise social media, the factors influencing their adoption within organizations, and the potential impacts of ESM on both individuals and organizations [21,31,32,33].

ESM allows managers to manage employees’ knowledge behaviors [34]. Moreover, some scholars have also started exploring knowledge behaviors within ESM, primarily focusing on a single knowledge behavior from a positive perspective [35,36,37]. Most of the research has predominantly focused on the positive side of ESM usage, such as knowledge sharing that can bring substantial benefits to businesses and unlock their true value [18,19,27,34,38]. It is indeed critical and significantly contributes to identifying and understanding the positive impact of ESM on organizations. However, many studies have failed to recognize the potential negative consequences of ESM usage, such as hindering employee productivity, impacting employee performance, and knowledge hiding [4,27,39]. These negative outcomes can also have implications for organizations [6,28,40]. Currently, the issue of knowledge hiding during the sharing process among employees within ESM has not received adequate attention from scholars and practical managers [12,36,37,41]. Nevertheless, knowledge hiding is equally critical because it hinders employee learning, productivity, and cooperation within the organization [37,38]. Hence, there remains a notable gap in the research regarding the negative perspectives surrounding knowledge behaviors in ESM. Moreover, it is insufficient to solely consider the positive or negative impacts of ESM usage in isolation. It necessitates considering both positive and negative outcomes into a theoretical framework for research to explain and predict the potential driving mechanisms behind different knowledge behaviors (such as knowledge sharing and knowledge hiding within ESM) [27,36,42,43].

In addition, the existing literature about knowledge behavior in ESM has examined the predictor’s factors either from the perspective of user motivation and self-characteristics (such as personal knowledge self-efficacy, self-presentation, identification, helping behavior) [16,27,44,45,46] or the social or organizational environmental factors (such as norms of reciprocity, social interaction, social trust, organizational support, and ESM affordances) [4,19,30,47,48]. There is a lack of comprehensive research frameworks from the ESM network environment and the employees’ psychological and cognitive perspectives to examine the influence mechanisms and boundary conditions [24,41,49,50]. However, both may play a significant role in developing knowledge interaction behavior on ESM platforms [51]. Social cognitive theory integrates individual and environmental factors, providing a comprehensive framework for understanding users’ online behavior [7,23,44,51]. This theory may be an ideal and comprehensive research framework for explaining the knowledge behaviors of ESM users.

Nevertheless, their emotional states may influence employees’ cognitive processes and subsequent behavioral decision making [52,53]. Positive emotions will facilitate employee beliefs and social media knowledge-sharing behavior but negative emotions will inhibit them [54,55,56]. Prior ESM and knowledge behavior research has been primarily based on cognitive and behavior models but rarely investigated the role of emotions [40,55,57,58]. Moreover, these studies have dealt with emotions as antecedents or mediation, and few have considered emotions as moderators [55,58,59,60]. Employees’ emotional states are essential components of work, and controlling emotions is considered a critical factor in determining employee success in the workplace [61,62,63]. Therefore, this study further assesses the possible moderation of emotions between employee social cognition and knowledge behavior in the ESM context.

This study aims to explore how social cognition and emotional factors influence employee knowledge-sharing and knowledge-hiding behaviors on the ESM platform. It contributes to the current ESM and knowledge management literature in the following aspects. Firstly, this study explored the paradoxically positive and negative knowledge behaviors—knowledge sharing and knowledge hiding—simultaneously within the context of ESM use in the workplace. Secondly, it revealed the mechanisms of how ESM context and employee individual cognition influence their knowledge-sharing and hiding behaviors. Thirdly, it highlighted the moderation role of employee emotion between social cognition and knowledge behaviors. This research will help managers gain a comprehensive and in-depth understanding of employee knowledge-interaction behavior, enabling targeted improvements and fostering effective knowledge exchange and management within organizations.

## 2. Theoretical Framework and Hypothesis

### 2.1. User Knowledge Sharing and Knowledge Hiding in Enterprise Social Media

#### 2.1.1. Knowledge-Sharing Behavior

Knowledge sharing is a prominent topic in knowledge management research, consistently attracting significant attention [15,18,48,64]. As various social media platforms have gained widespread usage, scholars have increasingly shifted their focus toward exploring knowledge-sharing behaviors within online communities and social media platforms [15,27,48,51]. As a web-based platform, ESM offers distinct characteristics that facilitate knowledge sharing. It allows employees to communicate online to contact anyone at anytime and anywhere through posting, editing, viewing, and sorting text and files in the organization [5,47,65]. The visualization features on ESM enable previously hidden employee communications to become visible to third parties [2,21,32]. Employees utilize ESM for activities such as sharing status updates, publishing opinions, engaging in project discussions, searching for experts, and exchanging knowledge and experience [2,51]. This enables the sharing of experts’ knowledge, experience, and skills among employees, fostering improved quality and efficiency in knowledge transfer [2,12]. ESM have the potential to combat organizational hierarchy and geographical boundaries by creating a platform that encourages the convergence of diverse employee ideas and facilitates the generation of fresh concepts [65]. They promote professional and complex knowledge sharing among employees, fostering a culture of collaboration and learning [15,48]. By effectively leveraging knowledge, employees can avoid redundant tasks, enhance their work efficiency, and accomplish more valuable tasks. This ultimately leads to faster and more comprehensive problem solving [2,15].

Previous research defines knowledge sharing in various contexts, such as online communities and social media. For instance, Razmerita et al. (2016) define knowledge sharing as a process wherein employees exchange tacit and explicit knowledge to generate new knowledge [30]. Connelly and Kelloway (2003) view knowledge sharing as an exchange process where platform members share information and knowledge with others, anticipating reciprocation in the future [66]. Knowledge sharing is also recognized as a dynamic process wherein users are willing and able to freely discuss and exchange their knowledge with others on a platform. Knowledge exchange enhances knowledge’s value and effectiveness, leading to positive outcomes [67]. Kwahk and Park (2016) define knowledge sharing as the behavior of individuals disseminating knowledge to other members within an organization [51]. Obrenovic and Du et al. (2022) deem knowledge sharing a voluntary exchange or transmission of knowledge, such as opinions, ideas, theories, and principles between individuals and organizations [19]. Therefore, considering the specific characteristics of ESM and the definition of knowledge sharing put forth by previous scholars, this study defines knowledge sharing as the process through which employees exchange and share their knowledge with other members within ESM, ultimately leading to higher-value knowledge.

#### 2.1.2. Knowledge-Hiding Behavior

The concept of knowledge hiding, introduced by Connelly and Zweig et al. [68] in 2012, has gained considerable attention from scholars in knowledge management in recent years. Knowledge hiding not only hampers the smooth flow of knowledge within ESM but also undermines employees’ creativity on these platforms, significantly diminishing knowledge creation and sharing [69]. Knowledge hiding is a prevalent individual behavior phenomenon within organizations [35,38,70,71]. Especially in the context of ESM usage, knowledge hiding is also prevalent [36,37].

Research indicates that as the costs and perceived risks associated with knowledge sharing increase, individuals tend to reduce their knowledge-sharing behavior and lean toward hiding their efforts [42,68]. Situational factors, complexity, and the difficulty of articulating knowledge all contribute to individuals’ inclination to accept knowledge shared by others while being hesitant to disclose their knowledge fully [69,70]. In the absence of pressure or triggers to engage in knowledge sharing, individuals often choose to withhold their knowledge, which is known as knowledge hiding.

Connelly defines knowledge hiding as a behavior in which individuals deliberately withhold or intentionally conceal knowledge requested by their colleagues within an organization [68]. Subsequently, based on Connelly’s definition, related scholars unanimously defined knowledge hiding as a deliberate attempt by employees to withhold or conceal knowledge from their fellow colleagues when related knowledge and information was requested. It is different from partial knowledge sharing or lack of knowledge sharing [72,73], and they may represent the opposite sides of the same continuum [42]. For example, lack of knowledge sharing occurs when employees genuinely do not know this knowledge, rather than intentionally hiding it [73]. Partial knowledge sharing occurs when some knowledge is shared, but the shared knowledge does not provide all the crucial information requested by the knowledge seeker. Additionally, partial knowledge sharing may result from inadequate absorption by the recipients rather than being an intentional act of not sharing [72]. Nevertheless, knowledge hiding must include both request and intention [35,37]. On the one hand, it emphasizes that knowledge hiding occurs in situations where there is someone to inquire or ask questions. On the other hand, it emphasizes the individual’s deliberate effort to conceal or withhold relevant knowledge, such as individuals exerting less effort or not giving their best when performing work-related tasks [41] and contributing less knowledge to others in the organization than they are capable of providing [24]. Therefore, it poses greater harm to organizations compared to a lack of knowledge sharing and partial knowledge sharing.

This article integrates the definitions of knowledge hiding provided by previous scholars to reach a definition of knowledge hiding on ESM as follows: deliberate behavior in employees, driven by personal interests, colleague relationships, or the sensitivity of knowledge and information, with the intention of withholding certain information or knowledge when faced with inquiries from other colleagues, rather than fully sharing it.

### 2.2. The Impact of Social Cognition on Knowledge Sharing and Knowledge Hiding

#### 2.2.1. Social Cognitive Theory

Social cognitive theory (SCT), proposed by Bandura in 1986, is widely applied to understand and predict individual motivations and behaviors in various contexts [74]. Social cognition theory states that individual behavior is controlled and determined by the social network environment, as well as by personal expectations and beliefs, which are the result of the interaction between the social environment and personal cognitive factors [23,72,75,76]. In other words, a specific behavioral pattern of social media users is influenced by their surrounding environment and their intrinsic cognition [51]. Hence, this theory provides an “environment-cognition-behavior” framework for explaining and predicting individual behaviors [44,64,76].

Previous IS research has extensively utilized this theory to study the adoption and acceptance of information technology, the use of information systems by users, computer training, and Internet behavior [77,78]. These studies have demonstrated the significant role of individual user’s cognitive factors, such as computer self-efficacy and outcome expectancy, in the adoption and use of information systems (IS) [77]. Currently, scholars have expanded the application of this theory to online communities and social media to explore user behaviors and knowledge-related activities [46,51,64]. It has been found that social cognitive theory provides a strong explanatory framework for understanding individual user behaviors and knowledge activities (knowledge sharing, knowledge creativity, knowledge contribution) within online communities and on social media platforms [7,51,79].

However, the application of social cognitive theory in the research of ESM is limited at present [7,76], and it has overlooked the influence of the social network environment [6,51,56]. In particular, there need to be more scholars who explain and predict the knowledge behavior of employees in ESM from the perspective of social cognitive theory [7,44,76]. ESM is the application of social media within an organization. The important user behaviors in ESM, such as knowledge interaction, may be the result of the joint action of ESM context factors and individual cognitive factors [7,51,76]. It is important to understand why employees are willing to invest their valuable time and effort in knowledge sharing with other members on these ESM platforms from both the ESM context and personal user cognition perspectives [43,44,75]. Therefore, social cognitive theory may also be applicable to the context of ESM, providing a possible theoretical framework for knowledge behaviors in ESM. To address the limitations of prior research, this paper adopts social cognitive theory as the theoretical framework to explore the driving impact of environmental factors and personal cognitive factors on employee knowledge-sharing and knowledge-hiding behaviors within ESM.

Previous research on online communities and social media has indicated that reciprocity and trust are important environmental and relational factors that drive users’ knowledge-related behaviors [78,80]. Reciprocity refers to the fair exchange of knowledge between users who recognize and accept each other [51,65,81]. By establishing connections and engaging in exchanges, users can acquire valuable resources for themselves, thereby maximizing their interests [78]. Trust, on the other hand, refers to the degree to which users believe in and are willing to take action based on the statements, actions, and decisions of others. Trust and a sense of responsibility among users on the platform are key factors in shaping their knowledge-related behaviors [27,64,70,80].

Furthermore, regarding users’ personal cognition, scholars have demonstrated that self-efficacy and outcome expectations exert significant influence on user behavior. Additionally, it has been pointed out that in voluntary knowledge-sharing situations, individuals who lack confidence in their ability to share knowledge are less likely to engage in such behavior [75]. Therefore, this study only considers the cognitive factor of outcome expectancy. Outcome expectancy refers to an individual’s anticipation of the possible outcomes that may result from implementing a particular behavior [64,70,82]. When employees use ESM, they make judgments about the expected impact and value of their knowledge-related behaviors [83], which, in turn, guide their subsequent knowledge behaviors.

#### 2.2.2. The Impact of Users’ Social Cognition on ESM Knowledge Sharing

Knowledge sharing in ESM refers to the behaviors in which employees exchange and share their knowledge on a platform [27,43,65]. This behavior plays a crucial role in facilitating the flow and transformation of knowledge, ultimately enhancing the effectiveness of knowledge sharing. However, the effectiveness of knowledge sharing can be influenced by various social cognition factors, such as reciprocity and trust.

Reciprocity is a norm that serves as the foundation for human communication and behavior [84]. It often serves as an internalized moral obligation to reciprocate the favors and efforts of others in a similar manner [84,85]. Previous research has indicated that it is considered an important factor in activities such as knowledge sharing and exchange among employees within organizations [46,51,85]. In the context of using ESM in the workplace, employees usually engage in activities such as reading, posting, asking questions, and seeking answers on ESM to facilitate information exchange and knowledge sharing [43,86]. Within ESM, knowledge sharing among employees is voluntary. However, knowledge sharing is not solely the result of a single person’s efforts but rather the outcome of interactive behaviors among users who possess the necessary knowledge [51]. Due to the limitations of their time, energy, and knowledge, employees generally expect favorable returns from their actions [81].

In addition, reciprocity is the perception of mutual assistance and fairness in knowledge-exchange behavior [51,87]. With more and more platform users believing in the existence of reciprocity, it is increasingly seen as a benefit of knowledge exchange [43,85,88]. Most users of ESM expect that their knowledge sharing will result in future returns [43,86]; that is, users who contribute more assistance tend to receive increased feedback and timely support from others when they are in need of help [89,90]. This virtuous circle continuously encourages knowledge contributors, fostering a strong sense of fairness and interactivity in the knowledge-exchange process. They believe that their efforts in ESM are valuable and rewarding, leading to a more positive attitude toward knowledge sharing [89] and increased participation in knowledge-sharing activities [81].

Hence, we hypothesized the following:

**H1:** 
*Reciprocity has a positive effect on knowledge-sharing behaviors in ESM.*


Trust is characterized as a set of specific beliefs regarding the reliability and integrity of an exchange relationship [30,51,65]. It has been widely discussed in the prior knowledge management literature and is considered another crucial factor in the network environment that influences employee cognition and knowledge-sharing behaviors [27,36,91].

In the process of using ESM, informal interactions among employees constitute a prominent feature. During these interactions, trust serves as the binding element that promotes cooperation between individuals [92,93,94]. When there is sufficient trust between the knowledge provider and the knowledge seeker, knowledge-sharing behaviors are more likely to occur [36]. Extensive research has been conducted on knowledge-sharing behaviors from the perspective of trust, demonstrating its essential role in fostering the development of social networks and creating an atmosphere conducive to knowledge sharing within an organization [27,95]. Furthermore, many scholars assert that trust creates and maintains exchange relationships and plays a vital role in high-quality knowledge sharing [27,96,97]. At the same time, lack of trust is considered the primary reason why users withhold their knowledge of the ESM platform [36].

Hence, we hypothesized the following:

**H2:** 
*Trust has a positive effect on knowledge-sharing behaviors in ESM.*


Outcome expectancy, which is a knowledge contributor’s belief in the potential outcomes of their knowledge-sharing behavior, may determine the occurrence of their knowledge-sharing behavior on the ESM platform (Bandura, 1986) [74]. Previous research in online communities and social media has demonstrated the positive impact of outcome expectancy on knowledge-sharing behaviors [64,75,82]. Drawing from expectancy theory, individuals are more likely to engage in knowledge sharing rather than knowledge hiding in ESM when the benefits of knowledge sharing outweigh the associated costs [98].

Previous research has indicated that individuals and employees may have both personal and social aspects of outcome expectations when they share knowledge on the ESM platform [32,51,99]. On the one hand, by using ESM for knowledge sharing, employees can gain expected individual benefits, such as self-presentation, exchanging ideas, seeking assistance, enhancing their platform status, and acquiring new knowledge and insights. On the other hand, using ESM for knowledge sharing can help employees expand their social network, develop new collaborations, earn respect, and achieve personal goals more efficiently, thereby achieving social effects. Thus, if employees have positive outcome expectations of knowledge sharing in ESM, they are more inclined to engage in sustained knowledge-sharing behaviors [82,83,100].

Hence, we hypothesized the following:

**H3:** 
*Outcome expectancy has a positive effect on knowledge-sharing behavior in ESM.*


#### 2.2.3. The Impact of Users’ Social Cognition on Knowledge Hiding in Enterprise Social Media

Knowledge hiding in ESM refers to the behavior in which employees choose not to share or withhold their own knowledge from others for various reasons. Prior studies found that a lack of reciprocity and trust and dissatisfaction with outcome expectancy may serve as the primary reasons why individuals opt for knowledge hiding rather than knowledge sharing [43,70]. Such behaviors significantly hamper the accumulation and development of knowledge within ESM.

Reciprocal norms influence the establishment and maintenance of knowledge-sharing behaviors. However, in the absence of reciprocity and without coercion or appropriate incentives, employees are more inclined to engage in knowledge hiding [24]. On the one hand, individuals who possess knowledge often perceive that sharing it will diminish its value while benefiting others. What is more, knowledge and context are complex and challenging to articulate, making it difficult to assess the effort exerted by an individual and the value of the shared knowledge. On the other hand, according to the norm of reciprocity theory, people should assist those who have helped them rather than harm them. However, this theory also acknowledges that individuals may seek retribution against those who have caused them harm [84]. In particular, individuals who have been rejected when seeking help in the past may retaliate through knowledge hiding. Previous studies have indicated that when a user is denied knowledge necessary for creativity, in turn, the same person is likely to reciprocate hidden knowledge back to the original knowledge hider. As a result, this behavior can subsequently hinder the creativity of the knowledge hider [35,38,101]. In other words, when employees hide knowledge, they will trigger a cycle of reciprocal distrust, in which colleagues become unwilling to share knowledge with them.

Thus, users who frequently experience disappointment from others are more likely to feel uneasy and distressed on a platform, making their behaviors more susceptible to others’ actions. These negative reciprocal experiences among employees in ESM can significantly increase knowledge-hiding behaviors [68]. On the contrary, in the presence of reciprocity, knowledge hiding decreases [70,101].

Hence, we hypothesized the following:

**H4:** 
*Reciprocity has a negative effect on knowledge-hiding behaviors in ESM.*


Trust is a crucial factor in individuals achieving high-quality knowledge exchanges during social interactions, and trust among employees in ESM serves as a significant foundation for fostering knowledge exchange [71,102]. Trust stems from the emotional connection between employees and serves as a critical psychological and emotional element in their willingness to maintain and contribute knowledge [65,97,103]. It plays a vital role in determining whether individuals choose to participate in or remain on ESM and whether they actively contribute knowledge to these platforms [24]. From an individual perspective, people who feel a sense of emotional attachment and connection are more likely to value the relationship, and a high level of trust generally reduces the uncertainty of knowledge sharing and the concern of losing valuable knowledge [27]. So, they are less inclined to hide knowledge. However, the absence of interpersonal trust can lead to the emergence of hiding behaviors [101]. The existing research has shown that individuals are unwilling to share their knowledge with others if there is a lack of trust [70,103]. And employees are more likely to hide knowledge from individuals they distrust. Moreover, the extent of knowledge hiding tends to increase in contexts of high distrust and competition [68,69,70,101]. Scholars have also pointed out that distrust and knowledge hiding interact and influence each other [36,65,97]. Knowledge hiding diminishes trust, and the decline in trust further exacerbates the occurrence of knowledge-hiding behaviors, ultimately forming a vicious cycle [68]. Hence, we hypothesized the following:

**H5:** 
*Trust has a negative effect on knowledge-hiding behaviors in ESM.*


Social cognitive theory highlights the idea that employees’ knowledge behaviors are primarily driven by their expectations and motivations for personal benefits, such as enjoyment and outcome expectancy [64]. Outcome expectations are in relation to the reward systems, which have a significant impact on individuals’ decisions to engage in knowledge sharing, whether or not [36]. When employees perceive that the effort they invest in ESM falls below or fails to meet their expectations, they are more likely to allocate less time and energy toward contributing to the platform [75]. Just as Alshahrani and Pennington (2021) indicate, there are three negative personal expected outcomes (distractions, privacy concerns, and time-consuming) and two negative social outcomes (distrust and plagiarism of their ideas) from the use of social media for knowledge sharing [82]. If employees expect that engaging in those sharing activities can lead to such negative outcomes, they will not share knowledge on the social media platform [64,82]. Therefore, during their usage of ESM, employees who have lower expectations regarding the outcomes of knowledge sharing are less inclined to engage in knowledge interactions and communication with other employees. Instead, they are more likely to choose silence and knowledge hiding. Hence, we hypothesized the following:

**H6:** 
*Outcome expectancy has a negative effect on knowledge-hiding behaviors in ESM.*


### 2.3. The Moderation of Emotions

The above discussion examines the antecedents of knowledge sharing and knowledge hiding among employees in ESM from a social cognitive perspective. However, these pathways may be influenced by employees’ emotional states [27,52,53].

Emotion as social information theory posits that emotions carry informative value, as the information expressed through emotional states aligns with individuals’ social judgments, cognitive processes, and user behaviors [53,104]. Emotion is a psychological state and a form of information. It not only provides individuals with information cues and signals about their environment but also feedback for individuals’ thoughts and cognitive tendencies [55,60,105], and then helps them make sense of complex situations and guides their behavior. It can stimulate cognitive processes and help people prioritize their behaviors by optimizing their regulatory responses to environmental demands [57]. Currently, an increasing number of scholars are recognizing the important role of emotions in organizational behavior research. For example, Beaudry and Pinsonneault (2010) indicated that emotions are an important driver of employee behavior, and they also play a significant role in employees’ understanding and utilization of emerging IT technologies and online platforms [57]. Wang and Zhou et al. (2015) developed a research model and found that positive emotion cues are a crucial enabler in driving users to instantly share information on microblogs [106]. Luqman, Zhang, and Kaur et al. (2023) demonstrated positive emotions as a connection element of colleagues, and teamwork can allow them to promote collaboration and knowledge sharing [55]. Masood, Zhang, and Ali et al. (2023) indicated that negative emotions embedded in the employees’ network might cause their social experiences greater harm than benefits in the workplace [27].

In ESM, due to the transparency of network connections and the visibility of communication content, employees tend to make emotional judgments and develop emotional inclinations toward the viewpoints and ideas expressed by their colleagues [55], thereby influencing their subsequent knowledge-related behaviors. Emotion as social information theory will help us better understand the differences in how different emotions can moderate employees making judgments about their ESM environment and personal cognition by evaluating the emotions expressed by others in social media interactions and subsequent knowledge-related decision making. In previous studies, it has also been treated as a moderating variable [60,105,107]. In light of this, this section aims to explore the moderating role of employee emotions in the aforementioned direct influence pathway.

Emotions are generally classified into two categories: positive emotions and negative emotions [108]. Positive emotions encompass feelings such as enjoyment, happiness, and satisfaction, while negative emotions include discontent, fear, anxiety, anger, and tension [54,57,109]. Previous research has also indicated that positive and negative emotions have different effects on beliefs in behavioral outcomes. Positive emotions facilitate such beliefs, while negative emotions inhibit their influence [110]. Therefore, within the context of ESM, different emotional states lead employees to experience distinct needs, judgments, and behavioral outcome expectations concerning the social network environment, resulting in differentiated behavioral manifestations.

Employees with positive emotions tend to have more positive judgments about their environment and see engaging in ESM as a pleasant experience with expected outcomes [54]. Employees with positive emotions are more likely to feel safe about their organizational environments, such as mutual trust among colleagues, reciprocity and mutual assistance, resources, and support from their teams and organization [39,55], which increase their trust level and make them more willing to invest effort in collaborative activities on ESM, and thereby promoting knowledge sharing [40,55]. Furthermore, some studies have found that positive emotional states encourage employees to pursue ideals, achievements, and rational and positive outcome expectancy [39,58,59]. They believe their efforts in the environment can bring them greater benefits, and they are more inclined to take the initiative in collaborating and seeking information [39], thus exhibiting a stronger enthusiasm for knowledge exchange and willingly sharing their thoughts, experience, and valuable information in their work [104,110]. Consequently, as positive emotions increase, employees’ social cognition factors (i.e., reciprocity, trust, and outcome expectancy) tend to exert a greater influence, stimulating employees’ engagement in ESM and their willingness to participate in knowledge sharing. Hence, we hypothesized the following:

**H7a:** 
*Positive emotions play a positive moderating role in the relationship between reciprocity and knowledge-sharing behaviors in ESM.*


**H7b:** 
*Positive emotions play a positive moderating role in the relationship between trust and knowledge-sharing behaviors in ESM.*


**H7c:** 
*Positive emotions play a positive moderating role in the relationship between outcome expectancy and knowledge-sharing behaviors in ESM.*


On the contrary, employees with negative emotions tend to have more extreme interpretations and judgments about their ESM environment, and these suppress their behavioral outcome expectations. Prior studies indicated that employees with higher levels of negative emotions perceive a lower likelihood of benefiting from reciprocity and trust in the environment. They distrust interpersonal relationships, and they usually feel that working conditions are unsafe [55]. Moreover, they often harbor strong dissatisfaction with their working environment and their work status [57]. Particularly, they are sensitive to the lack of reciprocity and trust in the work environment, which reduces employees’ cognitive resources and leads to counterproductive behaviors [55]. Several studies have found that employees with negative emotions are more likely to be concerned about sharing their information and knowledge. They tend to be more conservative and risk-averse [39,40].

In addition, employees with negative emotions often fail to meet their expectations despite their efforts [110]. This results in an inability to exhibit high levels of enthusiasm and motivation for work tasks, a lower willingness for self-development, and a reluctance to actively participate in knowledge exchange [36,106]. Instead, they are inclined to withhold the information and knowledge that they possess [54]. Employees with higher levels of negative emotions perceive a lower likelihood of benefiting from reciprocity and trust in the environment, which may inhibit knowledge [55]; thus, they tend to hide knowledge [57,111,112]. Therefore, we posit that negative emotions will weaken the negative impact of social cognition factors such as trust, reciprocity, and outcome expectancy on knowledge hiding.

**H8a:** 
*Negative emotions play a negative moderating role in the relationship between reciprocity and knowledge-hiding behaviors in ESM;*


**H8b:** 
*Negative emotions play a negative moderating role in the relationship between trust and knowledge-hiding behaviors in ESM;*


**H8c:** 
*Negative emotions play a negative moderating role in the relationship between outcome expectancy and knowledge-hiding behaviors in ESM.*


### 2.4. Control Variables

Control variables may have a potential impact on the dependent variables of knowledge sharing and knowledge hiding. For this research, we considered ‘age, gender, educational level, and frequency of use’ as the control variables since their role in ESM and knowledge management had been demonstrated in previous studies [27,58]. On this basis, we could deduce that employees with different ages, genders, educational levels, and frequencies of use may also be different in their purpose, time, resources, experience, and background, which may influence employee knowledge behaviors during the use of ESM.

Therefore, for this paper, the research model shown in Figure 1 was constructed.

## 3. Research Methods and Data Collection

### 3.1. Measurements

To ensure the reliability and validity of the construct measurement, the current study utilized existing scales for the measurement items whenever possible. These items were then adapted and supplemented based on the specific objectives of this article and the context of enterprise social media. In this study, a five-point Likert scale was developed, ranging from “completely disagree = 1” to “completely agree = 5”.

Based on this, a pilot test was administered to ensure its validity. The questionnaire was distributed to a sample of 10 professors and 50 employees who are currently studying in the MBA and MEM programs at our school and have extensive experience using ESM platforms. According to the preliminary analysis, we further modified and simplified the measurement items to form the final version of the questionnaire (shown in Appendix A).

Trust was measured by a four-item scale combined from Tan et al. (2000) and Carter et al. (2005), which addressed the level of trust that employees have in the reliability, security, and transparency of the platform and related activities [113,114]. The measurement of reciprocity drew on the works of Kankanhalli et al. (2005), which includes four questions that described the mutual assistance and fairness in employee knowledge interactions within enterprise social media [115]. The measure of outcome expectancy was developed by incorporating the scales of Hsu et al. (2004) and Constant (1994). It included four questions depicting the potential rewards and outcomes of using enterprise social media [98,116]. Inspired by the works of Cenfetelli (2004), Beaudry et al. (2010), and Watson (1988), a four-item scale of positive emotions and negative emotions was developed to measure psychological states and emotional tendencies [54,57,117]. Based on Bock’s study (2002), a four-question scale was developed that depicts the process of exchanging and sharing knowledge among employees on enterprise social media [83]. The knowledge-hiding scale was developed based on Connelly’s scale (2012), which includes four items to measure employees’ behaviors of intentionally concealing knowledge when faced with knowledge requests from others on enterprise social media [68].

### 3.2. Data Collection

Employees who worked in knowledge-intensive enterprises (such as IT companies and research institutions) were selected as the survey subjects. Knowledge-intensive enterprises are characterized by a high knowledge intensity and specialized knowledge with high requirements for knowledge management. These organizations were the earliest adopters of ESM for knowledge exchange and management in China and the employees may have more extensive experience of ESM use. This will help us gain a better understanding and investigation of employee knowledge behaviors in ESM.

This study adopted a convenience sampling method, utilizing MBA students, alumni resources, and strategic partner companies of the university to help us establish connections with some companies and conduct surveys in the Yangtze River Delta region. The data collection was carried out through web-based questionnaires by using a professional online survey platform, Wenjuanxing, in China.

The survey questionnaire consisted of two main parts. The first part included demographic questions aimed at collecting relevant information about the respondents, such as their gender, age, educational level, and usage of enterprise social media. The second part comprised measurement scales to assess the proposed model. Ultimately, a total of 276 electronic questionnaires were collected for this survey. After excluding invalid questionnaires with significant errors or inconsistent responses, a final sample of 240 valid questionnaires was obtained, resulting in an effective response rate of 86.9%. Descriptive statistical information regarding the sample is provided in Table 1.

## 4. Empirical Analysis and Hypothesis Testing

This research adopted the partial least squares (PLS) structural equation modeling (SEM) method to measure and validate hypothetical relationships and conceptual models. Smart PLS was utilized as the empirical data analysis tool [118]. The PLS-SEM method was chosen for several reasons. Firstly, it allows for the measurement of latent variables that are challenging to measure directly, thereby overcoming the measurement difficulties associated with the study variables. Secondly, PLS-SEM requires a smaller sample size compared to other SEM techniques, as long as the sample size is 10 times the maximum number of measurement items for any constructs (the minimum sample size for this study is 10 × 4 = 40 < 240) [119,120], so the sample size meets the minimum requirement for the PLS method. Thirdly, it is less sensitive to distributional assumptions. As a result, it was able to effectively measure and validate the formative and reflective indicators of the conceptual model.

### 4.1. Reliability and Validity Test

Firstly, this study examines the reliability and validity of the measurement model as a prerequisite for subsequent structural model testing. Table 2 shows the reliability and validity analysis of the measured variables in this study.

(1) Reliability test. The internal consistency coefficient (Cronbach’s α) values for all variables exceed 0.7, and the composite reliability (CR) values are more than 0.8, indicating acceptable construct reliability.

(2) Validity test. Firstly, as shown in Table 2, the average extraction variance (AVE) values of the validity measurement variables are all above 0.6, and the factor loads corresponding to all variables are close to 0.7 or above 0.7, which also shows that the measurement has good convergence validity. In addition, correlation coefficients were calculated among constructs such as reciprocity, trust, outcome expectancy, positive emotions, negative emotions, knowledge sharing, and knowledge hiding, and the square root of the AVE is placed on the diagonal of the correlation coefficient matrix for comparative analysis. As shown in Table 3, the square root of the AVE is larger than the correlation coefficients among all other constructs, indicating better discriminant validity in the construct measurement.

### 4.2. Common Method Variance

This study primarily employed procedural controls (such as anonymous responses and multiple questions) and Harmon’s single-factor analysis method to control and reduce the potential common method bias. All measurement indicators of first-order variables related to reciprocity, trust, outcome expectancy, positive emotions, negative emotions, knowledge sharing, and knowledge hiding, as mentioned in previous research, were included in the factor analysis. The analysis obtained five factors with eigenvalues larger than 1. The maximum variance explained by a single factor was 22.29%, which is less than 50% [121]. These results indicate that the influence of common method bias on this study was not significant.

### 4.3. Structural Model and Hypothesis Testing

The results of the reliability and validity testing of the aforementioned measurement model indicate that further testing of the structural equation model can be conducted. We use the values of the VIF, R^2^, Q^2,^ and goodness of fit (GoF) to verify the quality of the complete structural model.

As shown in Table 2, all the variance inflation factor (VIF) values of the constructs range from l.807 to 2.648, which meets the requirement of being below 3.33, indicating that there is no presence of collinearity [15]. As Figure 2 exhibits, the R^2^ values of knowledge sharing and knowledge hiding are 0.751 and 0.852, respectively, which demonstrate that the three aspects of social cognition factors and employee emotion factors together explain over 75% of the variance for knowledge sharing and over 85% of the variance for knowledge hiding. The results indicate that social cognition theory and emotion as social information theory have a certain explanatory power over knowledge sharing and knowledge hiding. Meanwhile, the cross-validated redundancy index Q^2^ value was calculated to examine the predictive accuracy of the structural model. The Q^2^ values for knowledge sharing and knowledge hiding are 0.705 and 0.824, higher than 0.5, confirming the high predictive relevance of the model [122]. In addition, the goodness of fit (GoF) of the structural model displays a figure of 0.554, larger than the recommended threshold of 0.36 [123], which further proves the quality of our structural model.

The result in Table 4 demonstrates that reciprocity (β = 0.621 ***, *t* = 8.450, *p* < 0.001) and outcome expectancy (β = 0.640 ***, *t* = 6.124, *p* < 0.001) both have a significant positive impact on knowledge sharing; thus, H1 and H3 pass the examination. However, trust has a significant negative impact on knowledge sharing (β = −0.382 ***, *t* = 3.650, *p* < 0.001), contrary to hypothesis H2, which is not supported by the findings. Reciprocity (β = −0.676 ***, *t* = 8.456, *p* < 0.001) and trust (β = −0.308 ***, *t* = 4.067, *p* < 0.001) both have a significant negative impact on knowledge hiding; that is, H4 and H5 are proven. However, the path coefficients for the influence of outcome expectancy (β = 0.047, *t* = 0.647, *p* > 0.05) on knowledge hiding are not significant; that is, H6 is not supported.

In order to examine the moderating effects of positive emotions and negative emotions, we further added interaction terms for positive (negative) emotions and reciprocity, positive (negative) emotions and trust, and positive (negative) emotions and outcome expectancy. As Figure 2 and Table 4 indicate that positive emotions do not significantly moderate the relationship between reciprocity and knowledge sharing (β = 0.014, *t* = 0.256, *p* > 0.05), H7a is not proven. At the same time, positive emotions negatively moderate the relationship between trust and knowledge sharing (β = −0.278 ***, *t* = 3.131, *p* < 0.01), which is contrary to H7b, so this hypothesis is not supported. Meanwhile, positive emotions play a role in strengthening the positive relationship between outcome expectancy and knowledge sharing (β = 0.200 *, *t* = 2.105, *p* < 0.05); that is, H7c is proven. The moderating effect is shown in Figure 3.

At the same time, the resulting values reveal that negative emotions negatively moderate the relationship between reciprocity and knowledge hiding (β = −0.178 ***, *t* = 3.885, *p* < 0.001), supporting H8a. The moderating effect is shown in Figure 4. The moderation effect of negative emotions on the relationship between trust and knowledge hiding is significantly negative (β = −0.125 *, *t* = 1.961, *p* < 0.05), leading to the acceptance of H8b. The moderating effect is shown in Figure 5. Negative emotions positively moderate the relationship between outcome expectancy and knowledge hiding (β = 0.146 *, *t* = 2.409, *p* < 0.05). Hence, H8c is not supported.

Figure 2 further shows the results of the control variables’ impacts on knowledge sharing and knowledge hiding are −0.032, 0.020, 0.059, and 0.022 and 0.007, −0.014, −0.041, and −0.073, respectively, indicating no significance. This suggests that variables such as age, gender, educational level, and frequency of use have not played important roles in this model.

## 5. Discussion

Drawing upon social cognition theory and emotion as social information theory, this study presents a comprehensive model that explores the influence mechanisms underlying employee knowledge-sharing and knowledge-hiding behavior in enterprise social media. It thoroughly examines the impact of the EMS environment and personal cognitive factors, including reciprocity, trust, and outcome expectancy on employee knowledge-sharing and knowledge-hiding behaviors. Moreover, it investigates the moderating role of positive and negative emotions in the aforementioned influence paths.

### 5.1. The Impact of Cognitive Factors on User Knowledge Behaviors in Enterprise Social Media

The above theoretical analysis and empirical results indicate that employees’ cognitive perception of the ESM environment and their personal cognition have a significant impact on their various knowledge behaviors.

Specifically, reciprocity and outcome expectancy have a positive impact on knowledge sharing, which aligns with the findings from previous research [51,82], such as Kwahk and Park (2016), who explored the impact of individual factors and social factors on knowledge-sharing activities, then on job performance. They concluded that the norm of reciprocity has a positive impact on knowledge sharing and personal performance in the ESM environment [51]. Nguyen, Malik, and Sharma (2021) combine the planned behavior theory and a motivational framework to study the online knowledge sharing of posters and lurkers in an organization. They found that reciprocity significantly affects posters’ knowledge-sharing intentions [43]. Hsu, Ju, and Yen et al. (2007), based on social cognitive theory, examined the impact of the environment and personal factors on knowledge sharing in virtual communities [64]. The results indicated that personal outcome expectations had a significant positive influence on knowledge sharing. The current results on knowledge-sharing behavior show that reciprocity and outcome expectancy serve as crucial driving factors for employees’ knowledge-sharing behavior, providing assurance for knowledge sharing and facilitating knowledge exchange among employees.

However, contrary to our expectations and prior research conclusions, trust exhibits a negative influence on knowledge sharing in the ESM context. Most scholars believe that trust, as an environmental factor, is beneficial for user participation on virtual platforms, as it helps individuals mitigate or reduce risks in various aspects such as economic benefits, recognition of abilities, respect, and status, thereby exerting a positive influence on knowledge sharing [27,36,65,66]. Unexpectedly, the current result did not obtain those similar results. But it was also found that trust had a poor explanatory for knowledge sharing. Especially, the effect of capability trust on knowledge sharing is negative, which is similar with our result [18,124]. One possible reason may be that the ESM is implemented within the company, and it is open communication in a virtual environment. And the use of ESM continuously accumulates trust among employees, enhancing the level of trust in the ESM environment, where there is a higher level of mutual trust among employees compared to public-orientated online communities and social media [27]. Highly trusting employees may be inclined to share less of their expertise because they anticipate that capable employees already possess the similar relevant knowledge [68].

Another possible explanation is that even in a high-trust ESM environment, there may still be competition (such as opportunities for honor, promotion, salary increase, etc.) among employees. According to communication visibility theory, ESM have effectively transformed the previously invisible nature of workplace communication [2,25,47], making communication between users more transparent. Communication interaction, network connectivity, and the content, opinions, and ideas expressed will be visible to other employees. Unless employees have sufficient self-confidence, they may otherwise fear losing their competitive advantage and may be especially unwilling to share tacit knowledge such as core knowledge and technical know-how with other colleagues, especially those with whom they have close business relationships and frequent interactions. Therefore, on an ESM platform, trust may be a necessary condition for employee knowledge sharing, but it may not have a positive impact on knowledge sharing in trust-based ESM contexts. A lack of trust may hinder people’s motivation to share knowledge with others, and highly trusting individuals may not necessarily be more willing to share knowledge than those with moderate or low levels of trust [78]. In a situation where professionals work together to accomplish tasks, there may be a sufficient level of trust. However, trust, at least in this case, may not adequately explain knowledge sharing [18,124].

Regarding the results on knowledge hiding, the above research findings demonstrate that reciprocity and trust have a significant negative impact on employee knowledge-hiding behaviors on ESM platforms. These findings align with previous research findings [36,101]. For example, Černe et al. (2014) found that knowledge hiding can trigger a cycle of reciprocal distrust, leading to colleagues being unwilling to share knowledge. However, in the context of reciprocal social exchange and a mastery climate, knowledge hiding is reduced [101]. Su (2021) used a social network approach to explore the influence of work and social relationships on knowledge sharing and knowledge hiding. The analysis results indicated that the trust network, interpersonal justice, and social communication have a negative impact on knowledge hiding [36]. This indicates that when employees establish certain reciprocity and trust norms within the organization, they are not inclined toward negative knowledge behaviors.

Outcome expectancy does not have a significant negative impact on knowledge hiding. This suggests that outcome expectancy may not be the primary influencing factor for knowledge hiding. One possible explanation is that ESM serves as a knowledge-sharing platform among internal employees, where employees are constrained by organizational power rules and routines while also experiencing the characteristics of a virtual community within the platform [70,82]. On the one hand, employees’ positive outcome expectations may enhance their engagement on the platform, leading to rewards from the organization and fostering knowledge sharing [64,82]. For example, in the process of employee participation in ESM, in addition to their own expectations of behavioral outcomes, the behavior of their colleagues also influences their behavior [51]. In order to avoid being labeled an “outsider”, and to gain recognition from leaders and colleagues, employees tend to align with their peers and engage in interactive communication within the ESM. However, out of self-interest and security considerations, employees selectively hide certain knowledge or provide vague answers. Through this discreet knowledge-hiding behavior, they ensure that they can obtain better benefits within the ESM. On the other hand, their positive outcome expectations may not necessarily reduce the possibility of hiding their perceived core techniques, valuable knowledge, and experience. This visualized communication carries the risk of exposing such knowledge to others [70]. Hence, it may be strategically and selectively hidden. Alshahrani’s and Pennington’s (2020) research indicate that individuals rationally pursue their own interests and carefully calculate the benefits of sharing knowledge by considering the types of resources that could potentially be exchanged [82]. In other words, positive outcome expectations can stimulate knowledge sharing, but they may not necessarily prevent a decrease in knowledge hiding [70].

### 5.2. The Moderating Effect of Positive Emotions and Negative Emotions

This study incorporates employee emotions into the context of ESM and examines the moderating effects of positive and negative emotions on the path from employees’ social cognition to knowledge behaviors. Very few studies have examined employee emotions in the ESM context and investigated its impact on employee knowledge-sharing and knowledge-hiding behavior.

The results indicate that positive emotions have a positive moderating effect on the path from outcome expectancy to knowledge sharing. This suggests that higher levels of positive emotions in employees are associated with more positive expectations and judgments about their behavioral outcomes [55,105,106], thereby promoting knowledge sharing on the ESM. Additionally, consistent with our predictions, negative emotions have a negative moderating effect on the path from reciprocity and trust to knowledge hiding, indicating that negative emotions weaken the negative impact of positive environments [27,54], such as reciprocity and trust, on knowledge hiding. It indicates that employees’ knowledge-hiding behaviors on the ESM are highly influenced by their negative emotions. Even if a positive environment with reciprocity rules and trust is established, employees’ negative emotions can still affect the effectiveness of their implementation. It is similar with the prior research result that negative emotions as a moderator have an amplifying effect [125]. This study advances the knowledge-behavior literature in the ESM context by verifying the role of emotions in employee social cognition and the effect on reducing knowledge-hiding behavior.

The moderating effects of positive emotions on the reciprocity–knowledge sharing link and trust–knowledge sharing link are not supported by empirical evidence. The results show that positive emotions did not significantly enhance the effect of reciprocity on knowledge sharing, but the path coefficient is a positive direction. Similarly, as demonstrated in the above discussion, trust did not bring significant positive effects to knowledge sharing in the ESM context whereas positive emotions can mitigate such an effect. Both of them indicate that positive emotions still play a positive role in fostering employees’ cognition and subsequent knowledge-sharing behaviors. Also, the moderating effects of negative emotions on the outcome expectancy–knowledge hiding link is not proved. This may be because negative emotions can influence individuals’ cognition and emotional state, thereby disrupting their perception and evaluation of outcome expectations [53,55,109]. Individuals consumed by negative emotions may increase the emotional burden, making it difficult for them to effectively process and utilize positive outcome expectations. In such cases, the situation where outcome expectations do not alleviate knowledge-hiding behaviors may be even worse.

## 6. Conclusions

### 6.1. Theoretical Contributions

This study contributes to the theoretical understanding and empirical examination of factors deemed significant in explaining how a user’s social cognition and emotion influence their different knowledge behaviors in the ESM context from a cognition and emotion perspective.

Firstly, this study contributes to the research on knowledge-sharing and knowledge-hiding behaviors in the ESM context. In contrast to previous studies on user knowledge behaviors in online communities and public-orientated social media, this paper extends the research to the internal enterprise social media context [4,9,20]. In addition, not only focusing on single and positive knowledge behavior [4,20,27], the current research examined the paradoxically positive and negative knowledge behaviors in ESM-based workplaces [36,38,39,126], which incorporated knowledge sharing and knowledge hiding within the same theoretical and empirical framework, and investigated both knowledge-sharing behaviors that bring real value to the use of ESM and knowledge-hiding behaviors that hinder the attainment of that value. The current results not only help us better understand positive knowledge behavior, but also aid in understanding negative knowledge behavior in the ESM context, which can potentially provide a reference for future research.

Secondly, this study expands the application of social cognitive theory to show how employees’ environmental and individual cognition influence their knowledge-sharing and knowledge-hiding behaviors in the virtual online context of ESM within the internal organizational setting. The existing research primarily focuses on applying this theory in technology adoption and usage and knowledge behaviors in online public communities [6,44,64,127], while the application of this theory in ESM within the workplace has not been thoroughly explored. From a social cognitive perspective, this study subdivides ESM context cognition into trust and reciprocity, considering outcome expectations as individual cognitive variables and exploring the differential impact of employees’ cognitive factors on the competing knowledge behavior (knowledge sharing and knowledge hiding). Our work provides empirical evidence to support the concept that employees’ context cognition and personal cognition are the critical factors that affect employee knowledge-sharing and knowledge-hiding behaviors [23,44,76], which provides an alternative theoretical research framework for future research on knowledge behaviors within the ESM context.

Thirdly, this study contributes to an increased understanding of the impact of employees’ emotional states on their social cognition and subsequent decision making regarding knowledge behaviors. It reveals a boundary condition of social cognitive theory about its influence on employees’ knowledge behaviors within the ESM context. The current study incorporates employees’ positive and negative emotions into the research framework of knowledge sharing and knowledge hiding, revealing the distinct influencing pathways of employee environmental cognition factors and individual cognitive factors on their knowledge-sharing and knowledge-hiding behaviors under different emotional states. This contributes to the existing literature by providing a more comprehensive picture and a compound theoretical framework to explain knowledge-sharing and knowledge-hiding behaviors on the ESM platform.

### 6.2. Practical Implications

The conclusions of this study also contribute to a better understanding of employee knowledge-sharing and knowledge-hiding behaviors on ESM platforms for businesses and the significant role played by various social cognition and emotion factors in those behaviors. This provides a theoretical basis for effectively utilizing knowledge resources within ESM and promoting ESM platforms’ sustainable and healthy development.

First of all, in the process of ESM usage, there are not only positive knowledge-sharing behaviors but also negative knowledge-hiding behaviors. Guiding employees from hiding to sharing knowledge is crucial for innovation, team cooperation, and organizational development [20,27,34,126].

During the process of designing, selecting, and implementing ESM, organizations need to build effective and comprehensive knowledge-interaction features. These features may include browsing, posting, replying, commenting, or engaging in interactions, recommending, sharing, private messaging, and more. Simultaneously, from a technical perspective, the platform should ensure that it offers characteristics such as protecting employee privacy and security, has a user-friendly interface, and is easy to use. Managers and leaders should actively participate in knowledge-sharing activities on the ESM platform and demonstrate the value of sharing knowledge. By setting a positive example, leaders can inspire employees to follow and create a culture of knowledge sharing within the organization.

Secondly, the research results indicate that employees’ cognition of trust and reciprocity in the ESM environment, and their personal outcome expectations, can help promote positive knowledge-sharing behaviors and inhibit knowledge-hiding behaviors. Research findings suggest that when an enterprise is designing ESM platform management policies and operational procedures, it is important to ensure the fairness and reasonableness of the policies. Firms should establish a positive and healthy ESM environment based on trust and reciprocity to encourage and stimulate employee knowledge-sharing behaviors while reducing and avoiding knowledge-hiding behaviors. On the one hand, firms need to pay attention to the reasonableness of their ESM platform management policies and operational procedures and ensure transparency in information access (for example, avoiding setting information-viewing permissions based on employee hierarchical differences). This fosters employee trust, ensures equality in communication among employees, and increases their motivation for knowledge sharing. On the other hand, when formulating platform regulations, firms should emphasize reciprocity and fairness. Platform operators should take employee feedback and suggestions seriously, establishing reward and penalty mechanisms to encourage knowledge sharing and interactions.

In addition, firms need to continuously improve the incentive and allocation systems for platform usage to ensure that employees can achieve their expected outcome through their own efforts. This is essential to effectively motivate employees, promoting continuous knowledge sharing within ESM, enhancing user engagement and platform activity, and contributing to the sustained and healthy development of ESM.

Thirdly, the research findings of the current study also suggest that firms should pay attention to employees’ emotional states. Because emotions can affect employees’ cognition and subsequent knowledge behavioral decisions, employee care and training should be implemented in the organization to cultivate positive emotions and foster employees’ self-development beliefs, thereby guiding positive knowledge behaviors. At the same time, offering tips and advice and blocking the insertion of bad advertisements and pop-ups on the platform can create a good emotional environment for employees on ESM and encourage them to engage in knowledge-sharing behaviors, rather than hiding knowledge.

### 6.3. Limitations and Future Research

Although this study has theoretical and practical significance, there are also some limitations. Firstly, regarding the research content, this article only explores the influence of reciprocity, trust, outcome expectancy, and other environmental and individual cognitive factors on employee knowledge-sharing and knowledge-hiding behaviors. Future research can deepen our understanding of employee knowledge-interaction behaviors in ESM by adopting new theoretical perspectives. Secondly, the data collection in this study was limited to a few selected companies, and the sample size needed to be bigger, which may introduce limitations. Subsequent research should expand the sample size to enhance the generalizability of the findings. Thirdly, the current study does not consider the influence of knowledge attributes that different factors may influence. The next step of research will distinguish between tacit and explicit knowledge, expanding the findings of this study. Lastly, relying solely on questionnaire surveys to collect data makes it difficult to fully capture employees’ emotional states during their participation in enterprise social media. Future research could combine web scraping and sentiment analysis methods to collect and analyze data, providing further validation and supplementation of the research results.

## Figures and Tables

**Figure 1 behavsci-14-00653-f001:**
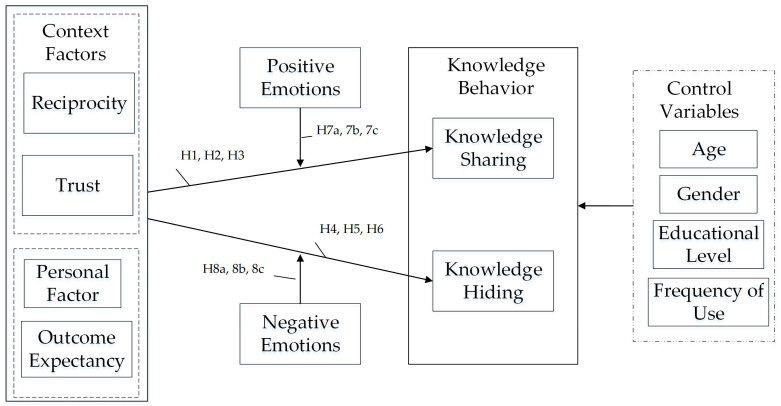
The framework model.

**Figure 2 behavsci-14-00653-f002:**
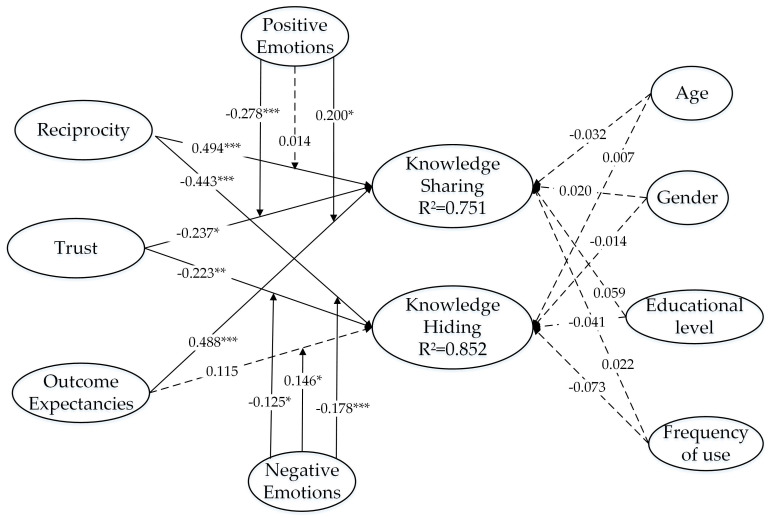
Test results of the model. Notes: ***, **, and *, respectively, denote *p* < 0.001, *p* < 0.01, and *p* < 0.05.

**Figure 3 behavsci-14-00653-f003:**
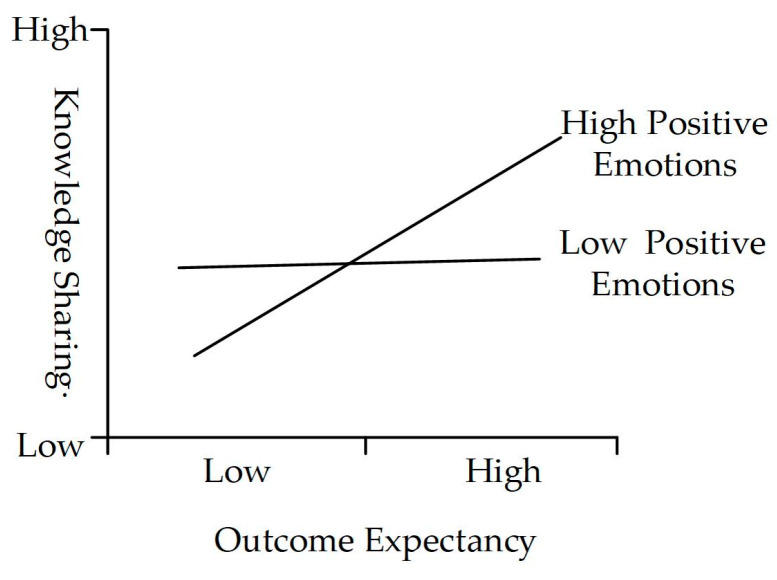
The moderation of positive emotions.

**Figure 4 behavsci-14-00653-f004:**
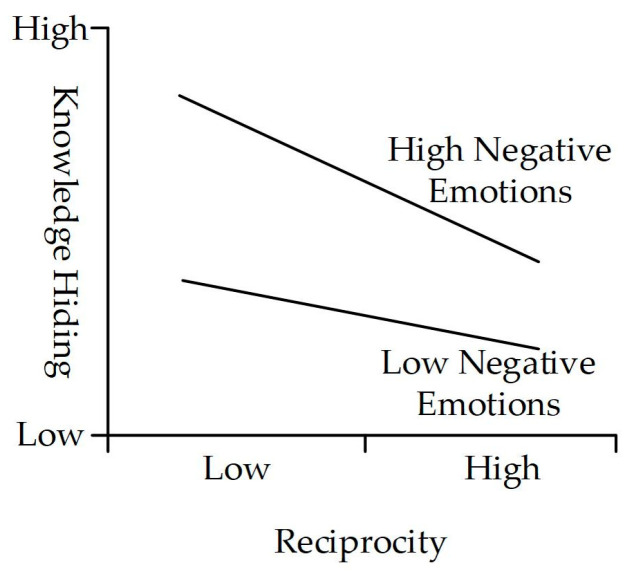
The moderation of negative emotions on Reciprocity-Knowledge hiding link.

**Figure 5 behavsci-14-00653-f005:**
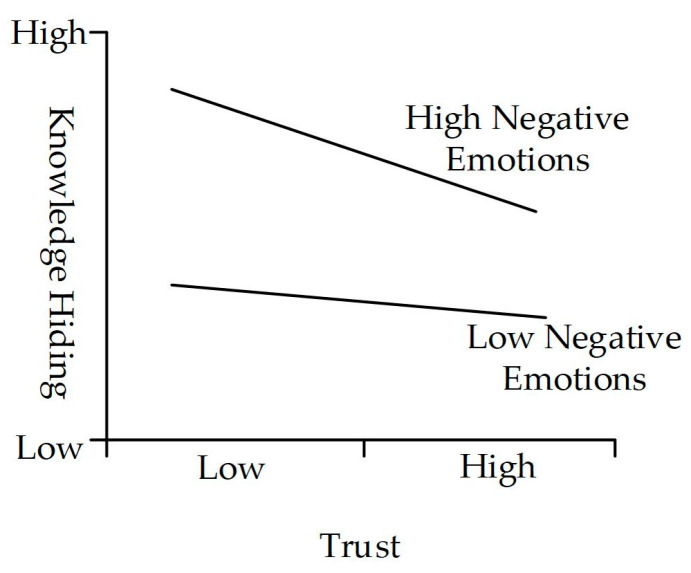
The moderation of negative emotions on Trust-Knowledge hiding link.

**Table 1 behavsci-14-00653-t001:** Descriptive statistics.

Project	Classification	Number of People	Percentage
Gender	Female	130	54.2%
	Male	110	45.8%
Age	Under 25	22	9.2%
	25–35	152	75.8%
	35–45	58	11.7%
	Over 45	8	3.3%
Education level	Below junior college	13	5.4%
	Undergraduate	109	45.4%
	Postgraduate	97	40.4%
	PhD	21	8.8%
Usage frequency of enterprise social media	Multiple times a day	54	22.5%
	Once a day	43	17.9%
	Once every two to three days	75	31.3%
	Once a week	60	25.0%
	Rarely used	8	3.3%

**Table 2 behavsci-14-00653-t002:** Reliability and validity analysis of the construct.

Construct	Measure	Factor Loading	Cronbach’s α	CR	AVE	VIF
Reciprocity	RE1	0.875	0.884	0.920	0.742	2.289
	RE2	0.853				
	RE3	0.852				
	RE4	0.865				
Trust	TR1	0.792	0.846	0.897	0.684	1.910
	TR2	0.848				
	TR3	0.805				
	TR4	0.862				
Outcome expectancy	OE1	0.876	0.903	0.932	0.774	2.596
	OE 2	0.880				
	OE 3	0.892				
	OE 4	0.871				
Positive emotions	PE1	0.828	0.834	0.889	0.667	1.807
	PE2	0.793				
	PE3	0.827				
	PE4	0.820				
Negative emotions	NE1	0.888	0.901	0.931	0.771	2.558
	NE2	0.871				
	NE3	0.878				
	NE4	0.874				
Knowledge sharing	KS1	0.897	0.902	0.931	0.773	2.648
	KS2	0.856				
	KS3	0.863				
	KS4	0.899				
Knowledge hiding	KH1	0.851	0.842	0.894	0.680	1.926
	KH2	0.809				
	KH3	0.773				
	KH4	0.862				

**Table 3 behavsci-14-00653-t003:** Correlation coefficients between constructs.

Construct Number	Reciprocity	Trust	Outcome Expectancy	Positive Emotions	Negative Emotions	Knowledge Sharing	Knowledge Hiding
Reciprocity	0.861						
Trust	0.628	0.827					
Outcome expectancy	0.568	0.826	0.880				
Positive emotions	0.843	0.588	0.481	0.817			
Negative emotions	−0.679	−0.479	−0.624	−0.596	0.878		
Knowledge sharing	0.762	0.552	0.698	0.648	−0.848	0.879	
Knowledge hiding	−0.847	−0.689	−0.592	−0.785	0.737	−0.800	0.824

Note: the diagonal of the correlation coefficient matrix is the square root of the AVE.

**Table 4 behavsci-14-00653-t004:** Analysis results of knowledge sharing and knowledge hiding.

Independent Variable	Dependent Variable			
	Direct Effect	Direct Effect	Moderate Effect	Moderate Effect
	Knowledge Sharing	Knowledge Hiding	Knowledge Sharing	Knowledge Hiding
Reciprocity	0.621 *** (8.450)	−0.676 *** (8.456)	0.494 *** (4.965)	−0.443 *** (5.406)
Trust	−0.382 *** (3.650)	−0.308 *** (4.067)	−0.237 * (2.048)	−0.223 ** (3.235)
Outcome expectancy	0.640 *** (6.124)	0.047 (0.647)	0.488 *** (3.939)	0.115 (1.752)
Positive emotions			0.149 (1.858)	
Positive emotions × Reciprocity			0.014 (0.256)	
Positive emotions × Trust			−0.278 ** (3.131)	
Positive emotions × Outcome expectancy			0.200 * (2.105)	
Negative emotions				0.387 *** (6.268)
Negative emotions × Reciprocity				−0.178 *** (3.885)
Negative emotions × Trust				−0.125 * (1.961)
Negative emotions × Outcome expectancy				0.146 * (2.409)
Gender	0.041 (0.628)	−0.027 (1.024)	0.020 (0.158)	−0.014 (0.838)
Age	−0.026 (0.744)	0.007 (0.207)	−0.032 (1.122)	0.007 (0.693)
Educational level	−0.037 (1.159)	0.024 (0.899)	0.059 (1.748)	−0.041 (1.432)
Using frequency	0.001 (0.007)	−0.039 (1.121)	0.022 (0.547)	−0.073 (0.103)
R^2^	0.725	0.757	0.751	0.852
Q^2^	0.703	0.735	0.705	0.824

Note: the *t*-values are enclosed in parentheses. ***, **, and *, respectively, denote *p* < 0.001, *p* < 0.01, and *p* < 0.05.

## Data Availability

The datasets used in the current study are available from the corresponding author upon reasonable request. The data are not publicly available due to the need to maintain the confidentiality of study participants.

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
