# Peer review of "Sharing or Hiding? Exploring the Influence of Social Cognition and Emotion on Employee Knowledge Behaviors within Enterprise Social Media"

_behavsci, 2024, doi:10.3390/bs14080653_

Round 1
Reviewer 1 Report
Comments and Suggestions for Authors
A paper on knowledge sharing, with standard data collection and analysis.
Some weaknesses include, small sample size od only 240.
Furthermore some of the variables such as reciprocity and expectancy have been well researched in the KS context. Some papers have already systematically researched gaps and factors studies related to KS. The following reference may help the authors understand the variables that were investigated in prior studies: Obrenovic, B., & Qin, Y. (2014) Understanding the concept of individual level knowledge sharing: A review of critical success factors. In Information and knowledge management (Vol. 4, No. 4, pp. 110-119).
The study does not distinguish between tacit and explicit knowledge, and those are influenced by different factors. So it should be added to the limitation section.
The hypotheses are not supported sufficiently
Overall study is comprehensive and well written, but originality is also questionable. Online communities have been studied before, so try to emphisize innovation of the study.
Comments on the Quality of English Languagecan proofread
Reviewer 2 Report
Comments and Suggestions for Authors
The article raises interesting and current issues.
The authors identified a research gap, built a model, and conducted analyses.
The number of respondents was very small. The research can be treated as a pilot study. Such general conclusions cannot be made based on the research conducted.
Reviewer 3 Report
Comments and Suggestions for Authors
Thank you for giving me this opportunity to review the manuscript entitled “Sharing or hiding? Exploring the influence of social cognition and emotion on employee knowledge behaviors within enterprise social media.”
1. Introduction
Some sentences needs citations. For example, the following sentences in introduction needs supporting citations. If can, please add citations as the authors check out the manuscript.
“Some scholars have made initial explorations into knowledge behaviors within ESM, primarily focusing on a single knowledge behavior from a positive perspective. Their studies have examined factors such as the organizational environment, technical availability, and employee motivation.”
2. Some sentences are the exactly same with the following article, He et al. (2022).
If can, please avoid similar expressions.
He et al. (2022). Empirical research on how social capital influence inter-organizational information systems value co-creation in China.
3. The purpose of this study
The purpose of this is not very clearly presented. The purpose and the contributions should be separately highlighted in the last paragraph in introduction.
4. Literature review
Literature review section failed to provide adequate support for a crucial idea, as it lacked citations from existing papers. This oversight suggests that the research may have overlooked important aspects or perspectives that are widely recognized in academic scholarship. Without proper validation from existing literature, the credibility of the argument presented in the literature may be questioned. It is necessary that key ideas are supported by previous research.
(pages, 3, 5, 6)
5. Control variables
Control variables are not supported by previous research.
6. The moderating role in emotions
I am not sure the moderating role in emotions because many studies demonstrated the mediating effect of emotions.
What are the differences between the moderating role and the mediating role?
Supportive literature should be sufficiently added in 2.3. the moderation of emotions.
7. Figure 1.
Please add hypotheses numbers (H1-8) in figure 1
8. Measurement
Please use the writing format in a consistent way on page 9.
9. What does it mean “Hsu et al., (2004) [62], Constant (1993) [47] and…” on page 9.
Constant is an author’s name?
10. Data collection
If the data are collected through both web-based and paper-based questionnaires, do the authors check out the differences?
The different methods for one research project may affect the representativeness of the data and even this study used a convenience sampling.
The data quality should be ensured.
Moreover, combining data from web-based and paper-based questionnaires requires careful attention to make sure compatibility and consistency.
12. Screening questions
More specifically describe how to select the respondents and add the explanations about screening questions.
11. P value
P=0.000 <0.001
I would recommend writing the results in the format commonly used in other academic papers.
12. Table
I would also recommend presenting the results on page 12 in a table
13. Discussion
Theoretical contributions
I would recommend comparing and contrasting the results to previous research.
Reviewer 4 Report
Comments and Suggestions for Authors
Dear Authors,
Although the interest of your paper, some improvements are needed, namely;
- Authors should clearly present and justify how they defined the target companies for the study. Once the companies were defined, how were potential collaborators and participants in the study identified? The authors used a convenience sample from among these collaborators. But how did they find them? This needs to be properly explained.
- The authors said that the questionnaires were administered online and on paper. However, they later only mention that they collected 276 online questionnaires. What about the paper questionnaires? Did no one respond? Please explain.
- The results discussion section requires greater depth and confrontation with supporting literature.
- In the conclusion section authors should, also, highlight the political implications of their study.
Comments on the Quality of English Language
English linguage is OK.
Round 2
Reviewer 1 Report
Comments and Suggestions for Authors
ok
Comments on the Quality of English Languageok
Author Response
|
Comments 1: [Comments and Suggestions for Authors is OK.] |
|
Response 1: Thank you for your recognition and support of the proposed modifications. |
|
|
|
Comments 2: [Comments on the Quality of English Language is OK.] |
|
Response 2: Thank you for your comments. |
Reviewer 3 Report
Comments and Suggestions for Authors
Thank you for your revision based on the comments and the authors’ response are presented specifically.
The color of the paper is green.
Results
I do not agree that display the results in Table and figure are redundant. The results should be presented.
I am not sure why the authors need to use SEM and stepwise regression model at the same time.
Author Response
|
Comments 1: [1. The color of the paper is green.] |
|
Response 1: Thanks for your comments. [The authors have removed the background color from the manuscript.] |
|
Comments 2: [ Results: I do not agree that display the results in Table and figure are redundant. The results should be presented. I am not sure why the authors need to use SEM and stepwise regression model at the same time.] Response 2: Thanks for your comment. [Yes, the authors may have misunderstood your previous comment about the result presenting. According to your suggestion, the authors have presented the data results in table 4 in manuscript. At the same time, the path coefficients for direct effects have been modified and adjusted according to the Table 4. The revised manuscript for this change can be found at page 14, paragraph 3 and 4, and page 15,16, table 4]
|
Round 3
Reviewer 3 Report
Comments and Suggestions for Authors
Thank you for your revision.
If the figures are larger, they would be better for readers.
Author Response
|
Comments 1: [Thank you for your revision. If the figures are larger, they would be better for readers.] |
|
Response 1: Thank you for your suggestion. [The authors have adjusted and enlarged all figures in the manuscript. The revised manuscript for this change can be found at page 11, figure 1, page 16, figure 2 and page 17, figure 3- figure 5] |